# Novel endosomolytic compounds enable highly potent delivery of antisense oligonucleotides

Jeremy P. Bost [1✉], Miina Ojansivu[2], Michael J. Munson [3], Emelie Wesén[4], Audrey Gallud[3,4], Dhanu Gupta [1], Oskar Gustafsson[1], Osama Saher [1,5], Julia Rädler[1], Stuart G. Higgins [6], Taavi Lehto[1,7], Margaret N. Holme [2], Anders Dahlén[8], Ola Engkvist [8], Per-Erik Strömstedt[8], Shalini Andersson [8], C. I. Edvard Smith [1], Molly M. Stevens [2,6], Elin K. Esbjörner [4], Anna Collén[3,9] & Samir El Andaloussi [1✉]

The therapeutic and research potentials of oligonucleotides (ONs) have been hampered in part by their inability to effectively escape endosomal compartments to reach their cytosolic and nuclear targets. Splice-switching ONs (SSOs) can be used with endosomolytic small molecule compounds to increase functional delivery. So far, development of these compounds has been hindered by a lack of high-resolution methods that can correlate SSO trafficking with SSO activity. Here we present in-depth characterization of two novel endosomolytic compounds by using a combination of microscopic and functional assays with high spatiotemporal resolution. This system allows the visualization of SSO trafficking, evaluation of endosomal membrane rupture, and quantitates SSO functional activity on a protein level in the presence of endosomolytic compounds. We confirm that the leakage of SSO into the cytosol occurs in parallel with the physical engorgement of LAMP1-positive late endosomes and lysosomes. We conclude that the new compounds interfere with SSO trafficking to the LAMP1-positive endosomal compartments while inducing endosomal membrane rupture and concurrent ON escape into the cytosol. The efficacy of these compounds advocates their use as novel, potent, and quick-acting transfection reagents for antisense ONs.

[1] Department of Laboratory Medicine, Karolinska Institutet, Huddinge 14157, Sweden. [2] Department of Medical Biochemistry and Biophysics, Karolinska Institutet, Stockholm 17177, Sweden. [3] Advanced Drug Delivery, Pharmaceutical Sciences, Biopharmaceutical R&D, AstraZeneca, Mölndal 43150, Sweden. [4] Department of Biology and Biological Engineering, Chalmers University of Technology, Gothenburg 41296, Sweden. [5] Department of Pharmaceutics and Industrial Pharmacy, Cairo University, Cairo 11562, Egypt. [6] Department of Materials, Department of Bioengineering, Institute of Biomedical Engineering, Imperial College London, London SW7 2AZ, United Kingdom. [7] Institute of Technology, University of Tartu, Nooruse 1, Tartu 50411, Estonia. [8] Discovery Sciences, R&D, AstraZeneca, Gothenburg 43150, Sweden. [9] Projects, Research and Early Development, Cardiovascular, Renal and Metabolism, Biopharmaceuticals R&D, AstraZeneca, Gothenburg 43150, Sweden. ✉email: jeremy.bost@ki.se; samir.el-andaloussi@ki.se

Nucleic acid-based therapeutics such as oligonucleotides (ONs) are gaining attention as a promising approach to interfere with gene expression across a range of pathologies[1]. One subset, termed splice-switching ONs (SSOs), are short (15–21 base pairs), synthetic, antisense ONs that have the ability to modulate the assembly of the splicing machinery to pre-mRNA[2]. SSOs can be designed to mask splice regulatory elements, leading to the inclusion or exclusion of introns and exons in mRNA. The resultant proteins may then have restored or inhibited function. It is estimated that up to 70% of human genes undergo alternative splicing, and up to 50% of human genetic diseases arise from mutations that affect splicing[3].

One of the largest hindrances to ONs' widespread use lies in their inability to escape endosomal compartments to reach the cytosol or nucleus of their target cells in sufficient concentrations[4]. In certain cases, the accumulation of ONs in the endosomal compartments and the slow release of the ONs from these compartments can be advantageous, as has been demonstrated with GalNAc-conjugated oligos recently[5]. However, GalNAc may be a special case as the asialoglycoprotein receptor is one of the fastest recycled receptors. For naked ONs, the release of ONs from the endosomal compartments is critical to achieve a fast cellular response. Hence, many of the recent developments in the field focus on increasing cellular uptake and endosomal release via chemical conjugation to ligands or encapsulation in various synthetic nanoparticles[6,7]. It was long considered a prerequisite to use nano-carriers in order to facilitate cellular uptake of polyanionic macromolecules such as SSOs. However, in the early 2010s, Stein et al. demonstrated that single-stranded ONs, such as gapmers, are in fact spontaneously endocytosed by cells, in the absence of carriers, by a process referred to as gymnosis[8]. In addition to gapmers, we and others have shown that SSOs of different chemistries can be gymnotically delivered to affect alternative splicing[9–11]. For gymnotic uptake to be active, relatively high concentrations of ONs are needed. In light of this, a third strategy can be exploited where endosomolytic small-molecule compounds (SMCs) are used to release gymnotically delivered ONs that otherwise predominantly accumulate in endosomes.

Endosomolytic SMCs can include cationic amphiphilic drugs (CADs). CADs induce endosomal membrane destabilization by buffering the lumen of endosomes as the luminal pH decreases during endosomal maturation. The buffering of luminal pH occurs quickly and can be reversible with proper dosing[12]. This buffering leads to an increase of luminal osmotic pressure, engorging the endosome and triggering membrane rupture, ultimately allowing the endosomal cargo to leak into the cytosol. CADs have been widely used to enhance the activity of ON-containing nanoparticles by promoting robust endosomal release[13,14]. Historically, the CAD most commonly used to facilitate ON-nanoparticle delivery is chloroquine[15]. Although displaying great potency in vitro, low efficacy requires that concentrations in the high micromolar range are typically used[16]. For reference, chloroquine induces leakage between 40–100 μM[17]. Juliano et al. has identified several families of compounds, including a compound UNC2383 which is capable of boosting ON activity by inducing ON escape from late endosomes at concentrations of 5–30 μM to boost ON activity[16,18]. Overall, there has been limited success to identify compounds with sufficient benefit-risk ratio to utilize for clinical use.

The field has acknowledged the need for widely applicable, sensitive characterization strategies to explore and optimize the use of SMCs for functional ON delivery, with novel imaging modalities at the forefront of this development[19]. For example, recent work has utilized live-cell microscopic assays which visualizes the recruitment of galectin-8 and galectin-9 (GAL8 and GAL9, respectively) to damaged endosomal membranes, and they have further been able to quantitate the galectin response in the presence of CADs[20–22]. In addition, the ability to visualize the mechanisms driving ON escape from endosomes by utilizing super-resolution microscopy is proving integral to understanding how SMCs work in the context of ON delivery[23].

This study aims to characterize a novel class of SMCs with a combination of functional and microscopic techniques. Compounds were selected from the AstraZeneca small-molecule compound library and assessed for their abilities to increase ON activity and to induce endosomal membrane rupture. In this report, we demonstrate that the splice-switching activity of SSOs occurs sequentially to this induced endosomal leakage, which can be greatly enhanced with optimized SMC co-treatment. The levels of splice switching correlate to observations obtained from quantitative analyses of the endosomal structure utilizing super-resolution fluorescence microscopy. We then demonstrate how functional SSO delivery occurs in a time and concentration-dependent manner correlating with endosomal rupture in a previously established microscopic GAL9 based assay[22]. Furthermore, we demonstrate the use of live-cell SSO-endosomal tracking microscopy to quantitate the colocalization of SSOs to various endosomal compartments over time. This combination of novel methods allows us to attain functional and mechanistic insight of SMC-mediated endosomal release of SSO. Importantly, we identify a novel compound that increases ON activity with greater efficacy than existing compounds and even promotes ON-mediated splice switching more efficiently than lipofection.

## Results

**Characterization of a series of novel endosomolytic small-molecule compounds (SMCs).** We aimed to characterize a novel endosomolytic compound that exhibits higher transfection efficiency/ability to induce endosomal escape of ONs than current compounds such as chloroquine that are used at high micromolar concentrations. While there exist several other approaches to increase ON activity such as bioconjugation to various delivery ligands or nanoparticle encapsulation, endosomolytic compounds offer the advantage that they can be utilized as a tool for evaluating ON efficacy in a screening setup due to the ease and simplicity of the co-treatment (there are no chemical conjugation steps, complexation, or particle synthesis steps needed). In addition, traditional lipid-based transfection methods and reagents, such as lipofectamine 2000 (LF2000) are limited by the cationic lipid's relatively high cytotoxicity[24,25]. The criteria for a suitable compound were: the compound (i) must be efficacious in a low-micromolar range; (ii) should only interfere with the normal cell trafficking machinery to a minimal extent until the point when leakage is induced; and (iii) should not induce plasma membrane damage or toxicity which irreversibly impedes cell proliferation or results in cell death.

In this study, compounds aimed for kinase inhibition were assessed for their ability to increase levels of a luciferase splice variant by simple co-treatment with the SSO in a HeLa_Luc705 reporter cell line. With the SSO luciferase variant reporter model, luciferase protein is produced as a result of functional SSO delivery and SSO activity can be quickly and sensitively quantitated[26]. Three of the selected compounds displayed strong functional activity at 5 μM, including the previously published endosomolytic compound UNC2383 (here CMP01) which was used as a positive control[18]. The two other compounds with increased SSO activity were CMP05 and CMP07 (structures of the functional compounds chosen to proceed with characterization experiments are shown in Fig. 1a); only CMP05 had significantly higher efficacy than CMP01 at the 5 μM dose and

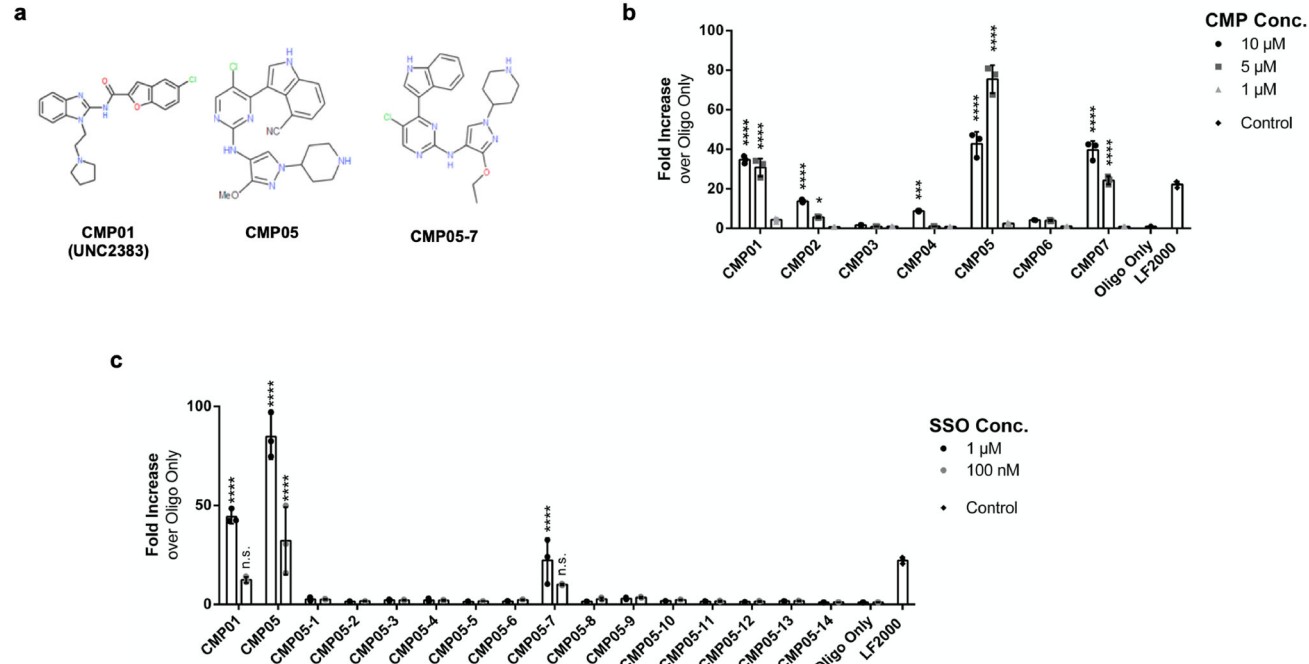

**Fig. 1 Assessing endosomolytic compounds for their ability to increase functional SSO activity. a** Chemical structures of CMP01, CMP05, and CMP05-7. **b** Functional SSO activity during co-treatment with chosen compounds. HeLa_Luc705 cells were pre-treated with 1 μM SSO, and then co-treated with 10 μM, 5 μM, or 1 μM of the compound for 2 h . Cells were then rinsed and incubated for 4 h in growth media. Data come from $N = 3$ experiments, each with $n = 3$ biological replicates. Values are reported as fold increase over activity from 1 μM naked SSO. Significance was determined (excluding LF2000) by two-way ANOVA followed by Dunnett's posttest to the oligo only control where ****$P < 0.0001$, ***$P < 0.001$, *$P < 0.05$, and n.s = not significant. **c** Functional SSO activity during co-treatment with CMP05 analogs. HeLa_Luc705 cells were exposed to 5 μM of the compound and either 1 μM SSO or 100 nM SSO for 2 h. Cells were then rinsed and incubated for 4 h in growth media. Values are reported as fold increase over activity from 30 h treatment with 1 μM naked SSO. Data comes from $n = 3$ experiments. Significance was determined (excluding LF2000) by two-way ANOVA followed by Dunnett's post test to the oligo only control where ****$P < 0.0001$ and n.s = not significant. Intraexperimental variability can contribute to discrepancies in fold increase between experiments.

was therefore chose to proceed with for an analog screen (Fig. 1b). At 10 μM, a decrease in activity was observed from CMP05 indicating that the optimal working concentration is below 10 μM.

Based on the chemical structure of CMP05, we designed 14 novel structural analogs (named CMP05-1, CMP05-2, etc.) which were synthesized and tested in the same HeLa_Luc705 SSO assay. The analogs were systematically developed to substitute functional groups of CMP05 for other groups of varying size and electronegativity, but the overall chemical scaffold was largely retained. These analogs were tested in co-treatment with 1 μM SSO and 100 nM SSO to identify compounds that enable the use of low-micromolar SSO concentrations (Fig. 1c). Most of the analogs did not induce an increase in activity. However, CMP05-7, which contains a methoxy to ethoxy substitution and deletion of the cyano group, displayed strong activity (Fig. 1c). We proceeded to use CMP05 and CMP05-7 in further characterization experiments to investigate the changes in SSO trafficking and to develop optimized characterization plans for the identified series of compounds.

Next, we sought to ensure that the increase in SSO activity was linked to endosomal rupture. Galectin-9 (GAL9) is a known marker of endosomal membrane disruption and can be utilized to quantitate endosomal rupture[22]. The initially selected 7 compounds and all 14 analogs of CMP05 were subjected to an mCherry-GAL9-recruitment assay to screen for the ability to induce endosomal rupture in both HeLa and HuH7 cell lines. Under normal conditions, mCherry-GAL9 is homogenously dispersed in the cytosol. When an endosomal leakage event is detected, mCherry-GAL9 will translocate to the damaged

endosome[19]. The resulting localization pattern of discrete puncta can be visualized and quantitated. HeLa_mCherry-Gal9 and HuH7_mCherry-Gal9 cells were subjected to a 10-point concentration dilution (0–15 μM) for 2 h and then fixed and imaged (Fig. 2a and Supplementary Fig. 1). Images were analyzed to quantify mCherry-GAL9 puncta and detect cell survival as determined by nuclear morphology (Fig. 2b). The quantitation of puncta reveals that CMP01 and CMP05 have a window between 1 and 8 μM in which Gal9 puncta formation increases, and at the lower end of this concentration window, nuclear morphology is conserved.

The assay was performed again, this time imaging several timepoints up to 4 h with 5 μM of each compound (supplementary Fig. 2a–c). This assay revealed that within 4 h, GAL9 recruitment is activated for several of the compounds. Only CMP05 induced GAL9 puncta formation within 15 min of compound treatment. CMP05 and CMP05-7 were chosen to move forward with the characterization experiments due to their efficacy at the 2-h timepoint and the fact that these two are structurally related, unlike CMP07.

**Characterization of endosomolytic compound dose and time-dependent effects.** To clarify how the compounds led to an increase in functional protein in the HeLa_Luc705 and HuH7_-Luc705 reporter models, we first sought to confirm the presence of splice-altered luciferase RNA in the treated cells. RT-PCR confirmed that the increase in functional luciferase production is the result of antisense SSO activity on the target Luc705 pre-mRNA in the nuclei of both HeLa and HuH7. Quantitation of the

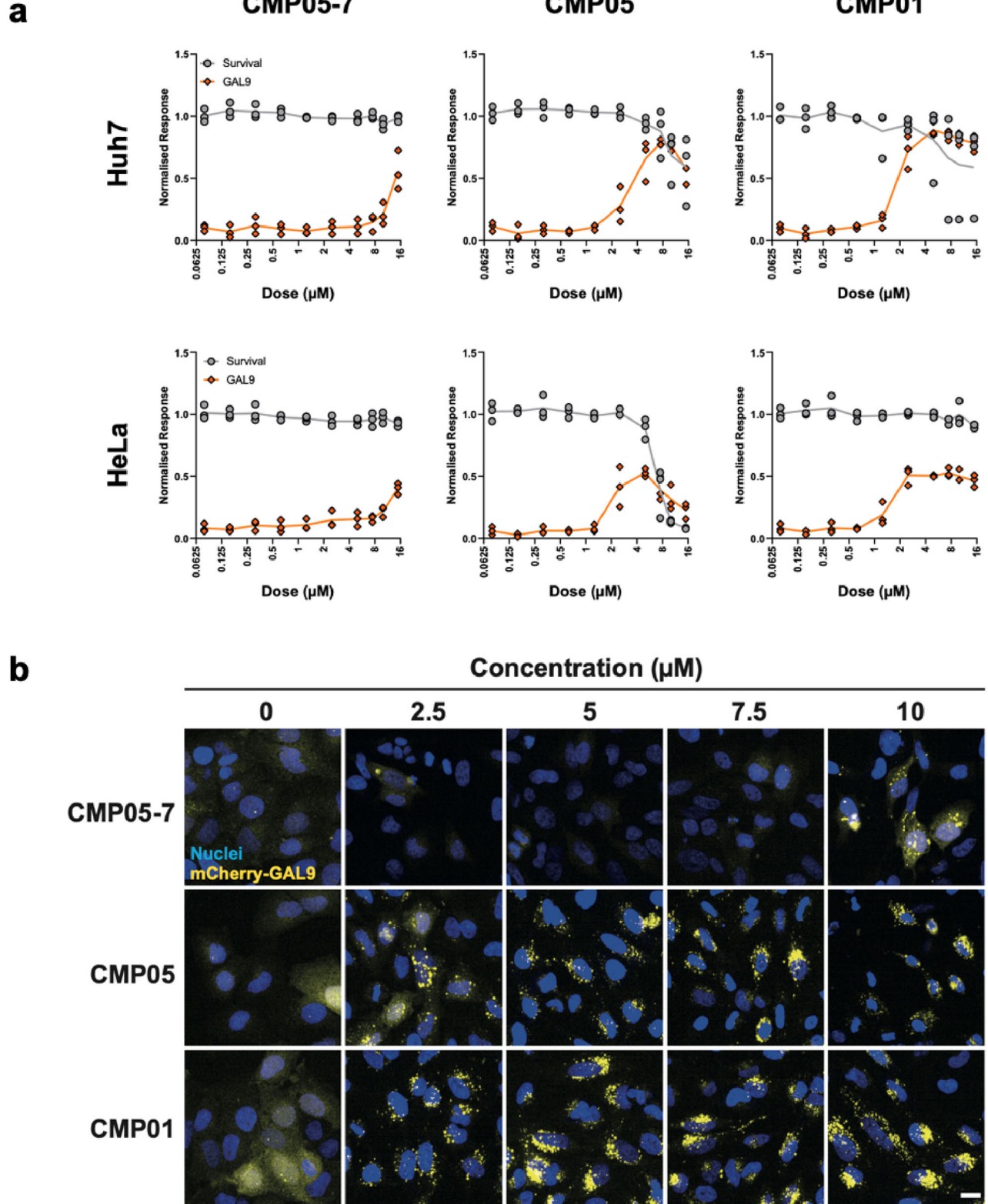

**Fig. 2 Screening endosomolytic compounds for Gal9-related endosomal rupture. a** HeLa_mCherry-Gal9 cells were treated with compound across 10-point dose titration series for 2 h and then stained for nuclei (Hoechst 33342, blue), fixed, and imaged. Representative images display the translocation of Gal9 (yellow) from cytosolic to punctate in a dose-dependent manner. Scale bar = 20 μm. **b** Image analysis reveals the narrow window in concentration between induction of endosomal rupture and cytotoxicity. Gal9 punctate (orange) reported as a percentage of max quantitation, normalized to nuclei signal. Cell survival (gray) reported as a percentage of all cells normalized to a DMSO control. HuH7 images are shown in Supplementary Fig. 1.

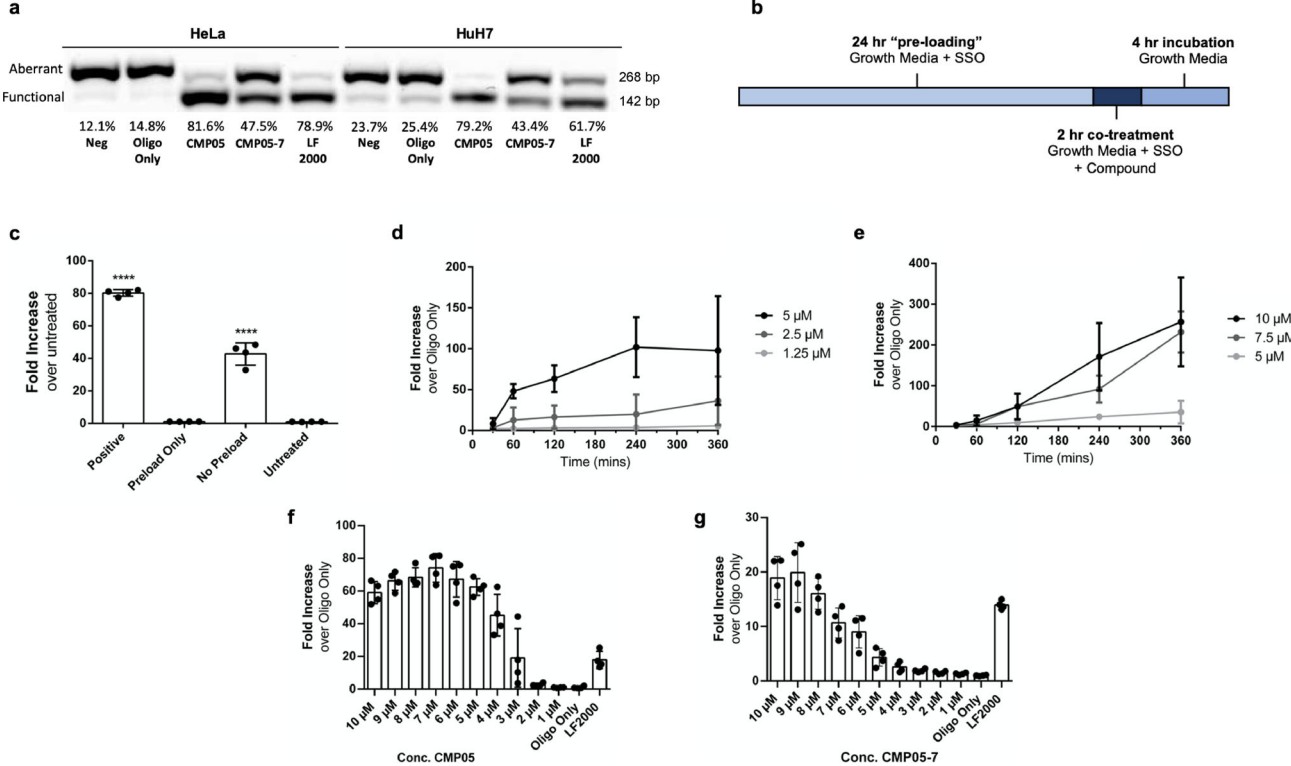

**Fig. 3 Characterization of endosomolytic compound dose and time-dependent effects. a** Representative RT-PCR with gel analysis used to detect aberrant or functional mRNA copies. Gel images were quantitated using Quantity One (BioRad) software. Values of functional mRNA are reported as a percentage of total mRNA. **b** Schematic of the treatment plan. Cells are seeded in DMEM + 10% FBS (growth media) with added SSO. After 24 h, endosomolytic compound diluted in growth media is added to the desired final concentration. After 2 h of co-treatment, media is removed, cells are washed with PBS, and then replaced in growth media for 4 h. Media is then removed and cells are lysed in accordance with the next experimental steps. **c** Functional SSO activity from different treatment plans of CMP05. The "positive" group underwent the previously described treatment. The"preload only" group was treated with 1 μM SSO for 24 h and then media was changed to growth media with 5 μM compound for 2 h. The"no-preload" group was treated with 1 μM SSO and 5 μM CMP05 simultaneously for 2 h. Values are reported as fold increase over untreated cells. The "Positive" and "No Preload" groups are each significantly higher than the "Untreated" and "Preload Only" groups ($P < 0.001$). Data come from $n = 4$ experiments. **d, e** HeLa_Luc705 cells were co-treated with 1 μM SSO and various concentrations of CMP05 (**d**) or CMP05-7 (**e**). Co-treatment duration lasted 30 min, 1 h, 2 h, 4 h, or 6 h followed by the same 4-h incubation step. Values are reported as fold increase over SSO only. Data come from minimum of $n = 3$ experiments (**d**) and $n = 3$ experiments (**e**). **f, g** HeLa_Luc705 cells were co-treated with 1 μM SSO and various concentrations of CMP05 (**f**) or CMP05-7 (**g**). Co-treatment duration lasted 2 h, followed by a 4-h incubation step. Data comes from $n = 4$ experiments. Intraexperimental variability can contribute to discrepancies in fold increase between experiments.

mRNA splice variants revealed increases in the percentage of alternatively-spliced mRNA (Fig. 3a) in the conditions treated with CMP05 and CMP05-7. Notably, treatment with CMP05 and SSO was shown to induce greater levels of splice switching than transfection with SSO using Lipofectamine 2000 (LF2000), a widely used transfection reagent, in both HeLa and HuH7 cells. One major limitation of LF2000 is that it must be complex with the SSO such that the ratio of LF2000 to SSO remains constant for optimal transfection efficiency. Due to this, the highest concentration of SSO which can be treated to cells in vitro is 200 nM. In contrast, the endosomolytic compounds and SSO work independently of each other, allowing use of SSO concentrations in the micromolar level.

We hypothesized that SSO and compound must be present in the cell media simultaneously for the CMP05 to be efficacious. The original treatment protocol involved a pre-treatment with SSO only for 24 h, followed by the addition of CMP05 for 2 h, followed by the removal of SSO and CMP05, and finally a 4-h protein translation period (Fig. 3b). If the pre-treatment step was increased to 48 h, the increases in SSO activity were even more pronounced (Supplementary Fig. 3a, b). Importantly, if SSO is only pre-treated, then removed when the compound is added,

then no activity was seen (Fig. 3c). Increased activity could also be observed even without the pre-treating step (Supplementary Fig. 3c, d). These results imply that it is necessary that cells are exposed to the compound and the SSO simultaneously, and therefore that the newly formed endosomes contain both in order to elicit a functional effect from the compound.

Using the Luc705 cell model, treatment of CMP05 and CMP05-7 was optimized in regard to both compound concentration and treatment duration (Fig. 3d–g). Co-treatment with CMP05 increased SSO activity up to 5 μM, leading to a 100-fold increase in activity, plateauing after 2 h. Above 5 μM, luciferase was detectable, but cells developed morphological changes (as noted in the survival curves generated in Fig. 2). We therefore decided the optimal CMP05-treatment plan in HeLa cells to be 5 μM for 2 h, and most of our following characterization experiments follow this plan. Further, we observed that CMP05–7 led to a 200-fold increase in SSO activity up to 6 h at 10 μM. The results demonstrate that the compounds promote SSO-mediated splice switching in a dose-dependent manner. In addition, the results from these functional experiments reveal that the SSO activity is occurring in a manner that mirrors the Gal9 puncta recruitment across the tested timepoints and doses.

This supports the hypothesis that these compounds enhance delivery efficiency by inducing endosomal rupture[19,20].

Next, we set out to investigate whether the compounds impact cellular plasma membrane integrity. Even though negligible toxicity was observed in the Gal9-imaging assays, we wanted to assure that the effects of the CMPs are not merely through destabilization of the plasma membrane. To this end, acute toxicity was assessed by using a lactate dehydrogenase (LDH) assay which detects the presence of this enzyme in the culture media after treatment of cells. After 2 h of treatment with either CMP05 or CMP05-7 (Supplementary Fig. 3e, f) we did not detect membrane leakage of LDH, further corroborating that the compounds act in the endolysosomal pathway.

In addition to HeLa- and Huh7-derived cell lines, we sought to deliver SSO to Neuro-2A (N2A), a neuroblastoma cell line, and U2-OS, an osteosarcoma cell line, with CMP05 and CMP05-7. Our group has previously generated N2A and U2-OS cell lines that express the 705 reporter construct[27]. The cells were treated with the optimized treatment timepoint plan at varying concentrations of CMP05 or CMP05-7 (Supplementary Fig. 4). Importantly, CMP05 and CMP07 were able to increase SSO activity in both additional cell lines. Peak activity was observed at different concentrations between cell lines and compounds, with U2-OS cells showing the highest activity with CMP05 at 2 μM, and N2A between 5 and 8 μM.

We next sought to confirm that the mechanism behind the compound-induced SSO activity was dependent on the acidification of the endosomes. As endosomes mature, their pH decreases as a result of protons being pumped into the endosomal lumen by H(+)-translocating ATPases. Bafilomycin is a well-characterized inhibitor of H(+)-translocating ATPases that others have utilized to demonstrate a delivery system's dependence on endosomal acidification[28]. In line with this, 2 μM bafilomycin co-treatment inhibited CMP05-induced SSO activity (Supplementary Fig. 5). To further investigate how the compounds influenced the endosomal network, we proceeded with several microscopy-based approaches.

**SMCs induce SSO leakage concurrently with endosomal swelling**. To visualize the effect of CMP05 on endosomal structure, we employed stochastic optical reconstruction microscopy (STORM), a super-resolution microscopy visualization technique which can reach an imaging resolution of about 20 nm (Fig. 4)[29].

HeLa and HuH7 cells were treated with 5 μM or 2.5 μM CMP05, respectively, for 2 h. Cells were then fixed and stained for late endosome-associated membrane protein 1 (LAMP1), a known marker of acidic endosomal/autophagy compartments, predominantly late endosomes and lysosomes[30]. Figure 4a and c shows representative STORM images of lysosomes in HeLa and Huh7 cells, respectively. Quantitation and characterization of the LAMP1-positive bodies in the STORM visualizations were performed via ilastik, an image-analysis software capable of using machine learning algorithms to segment and classify the desired structures (Supplementary Fig. 6). In HeLa cells, CMP05 treatment-induced swelling of the LAMP1-positive endosomes, increasing the median diameter of LAMP1 bodies from 0.24 to 0.32 μm, shown in Fig. 4b (Welch's $t$ test, $t = -10.43$, $df = 615.13$, $P = 1.5 \times 10^{-23}$). This effect was conserved in Huh7 cells, with an increase in the median diameter from 0.27 to 0.34 μm, shown in Fig. 4d (Welch's $t$ test, $t = -6.33$, $df = 502.95$, $P = 5.4 \times 10^{-10}$).

STORM was utilized further to investigate whether the compound-induced endolysosomal swelling was associated with SSO leakage into the cytosol. HeLa and HuH7 cells were pre-loaded with 1 μM of either Alexa-488-labeled SSO or Alexa568-labeled SSO (488-SSO and 568-SSO) and then co-treated with

CMP05 for 2 h. Cells were again fixed and stained for LAMP1 before imaging. In the presence of SSO, HeLa cells displayed the same LAMP1-body engorgement as seen in the Fig. 4. In the compound-untreated group, SSO predominantly displayed an endosomal localization pattern with a small amount of 488-SSO present in the cytosol (Fig. 5a). A high degree of colocalization between the 488-SSO and the LAMP1 could be noted in the untreated group. In the treated group, however, there was a near-complete loss of colocalization between LAMP1 and 488-SSO, easily observed in both the widefield and STORM images.

Huh7 cells displayed a similar pattern, however, it must be noted that the 488-SSO was not as efficiently escaped from the endosomes in response to compound treatment as in HeLa cells (Fig. 5b). Yet, the Huh7 images revealed that after treatment, 488-SSO had lost much of its colocalization with LAMP1.

One experimental concern was that the hydrophobicity of the fluorescent label on the SSOs could facilitate non-specific protein binding and affect their ability to escape the leaky LAMP1 bodies[31,32]. Therefore, these imaging experiments were repeated with 568-SSO which contains a fluorophore that is larger and more hydrophobic than 488-SSO[33]. The Alexa568-tag appeared to slightly inhibit the ability of the SSOs to fully escape from the endosomal compartments in HeLa, however, the majority of 568-SSO was still cytosolic after treatment (Supplementary Fig. 7a, b). These results indicate that although the fluorophore may have slight effects on the ON localization, the overall localization patterns strongly support the hypothesis.

**SMCs induce SSO leakage in a mCherry-GAL9-recruitment-dependent manner**. Next, HeLa and HuH7 cells were again subjected to the mCherry-GAL9-recruitment analysis, this time co-treating CMP05 with 1 μM 488-SSO. After preloading 488-SSO for 24 h, 488-positive structures that likely reflect endosomal localization could easily be detected. These 488-positive structures gradually decrease in number over 2 h under normal circumstances. This disappearance was greatly expedited during treatment with CMP05, indicating endosomal rupture and subsequent leakage of 488-SSO into the cytosol where it is too diffuse to detect with this microscopic platform (Fig. 6a). This rupture and the loss of 488-SSO puncta occurred in a time and concentration-dependent manner, consistent with earlier functional CMP05 findings (Fig. 6b). Simultaneously, the GAL9 re-localization resulted in the formation of mCherry-GAL9 puncta (Fig. 6c). The quantitation of GAL9 also revealed time and concentration-dependent responses, supporting the hypothesis that 488-SSO endosomal escape occurs in a manner coinciding with GAL9 recruitment.

**SMCs disrupt normal SSO trafficking through the endosomal network**. Having demonstrated a loss in SSO accumulation in LAMP1-positive vesicles, we next sought to determine whether SSO leakage was specific to these vesicles or if this loss of accumulation is present with other endolysosomal bodies. We therefore performed live-cell microscopy with three modified HuH7 cell lines that stably overexpressed fluorescently tagged endosomal markers: mRFP-RAB5 for early endosomes, mRFP-RAB7 for late endosomes, and LAMP1-RFP again for late endosomes and lysosomes. To enable analysis of intracellular SSO trafficking, these cells were pulse-incubated with 488-SSO and CMP05 for 15 min, washed to remove non-internalized SSO, and thereafter imaged over time during continuous incubation with CMP05 (Fig. 7a–c). Quantitative analysis revealed that CMP05 treatment leads to a decrease in the number of detected SSO-positive vesicles also following SSO pulse-incubation in all tested cell lines (Supplementary Fig. 8). This is consistent with recent findings

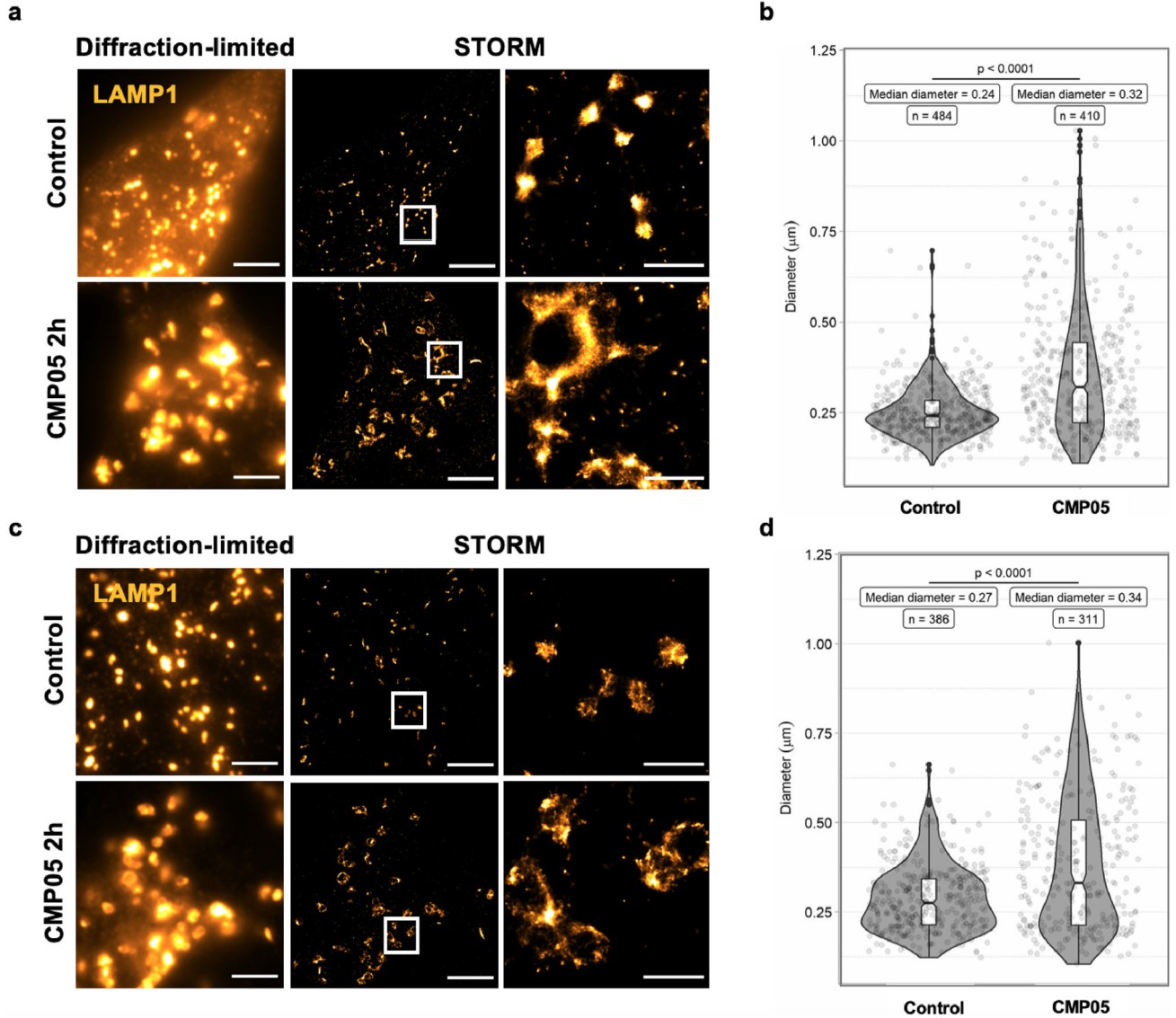

**Fig. 4 Visualization of lysosomes in control and CMP05-treated HeLa and HuH7 cells. a** Diffraction-limited and STORM imaging of LAMP1 immunostaining in HeLa cells after 2 h, 5 µM CMP05 treatment. Scale bars in full-size images are 5 µm and in the enlarged area 1 µm. Enlarged visual fields are identified by white boxes. **b** Quantification of the lysosome sizes in STORM images of the HeLa cells using ilastik software. Smoothed curves show the kernel density estimate for each population, overlaid with boxplot of quartiles and the underlying data. **c** Diffraction-limited and STORM imaging of LAMP1 immunostaining in HuH7 cells after 2 h, 2.5 µM CMP05 treatment. Scale bars in full-size images 5 µm and in the zoomed area are 1 µm. **d** Quantification of the lysosomes sizes in STORM images of the HuH7 cells using ilastik software. Smoothed curves show the kernel density estimate for each population, overlaid with boxplot of quartiles and the underlying data.

from Kondow-McConaghy et al. which demonstrate a decreased rate of endocytic uptake after treatment with endosomolytic compound[34]. Colocalization analysis between SSO-positive vesicles and the endolysosomal markers was performed in a particle-based manner using the ComDet plugin to ImageJ. Analysis reveals that under control conditions, colocalization remains steady between RAB5 and SSO, while colocalization with SSO increases with time in both RAB7 and LAMP1 (Fig. 7d). This implies that SSO passes through early endosomes then begins to accumulate in late endosomes and lysosomes. In the CMP05-treated samples, however, a steady loss of colocalization occurred in the RAB5 samples, followed by a lack of accumulation in late endosomes and lysosomes. Together with the STORM imaging data, these findings indicate that CMP05-induced SSO escape occurs in a range of endolysosomal vesicles, from early to late endosomes and lysosomes.

## Discussion

In recent years, much attention has been given to intracellular trafficking of ONs and it is known that the majority of internalized ON molecules are sequestered in the endolysosomal system, never reaching their nuclear or cytosolic targets to have a functional effect. The endolysosomal pathway is a highly dynamic system that involves continuous membrane fusion and division events. Thus, there is an opportunity to induce membrane disruption which increases functional ON delivery.

Attention across the field has turned to endosomolytic small molecules considering their potential to increase the activity of ONs regardless of the ON chemistry or target. However, relatively little work has been done to elucidate with certainty the altered-ON trafficking patterns.

In this context, we sought to identify previously uncharacterized small molecules which increase ON activity by influencing

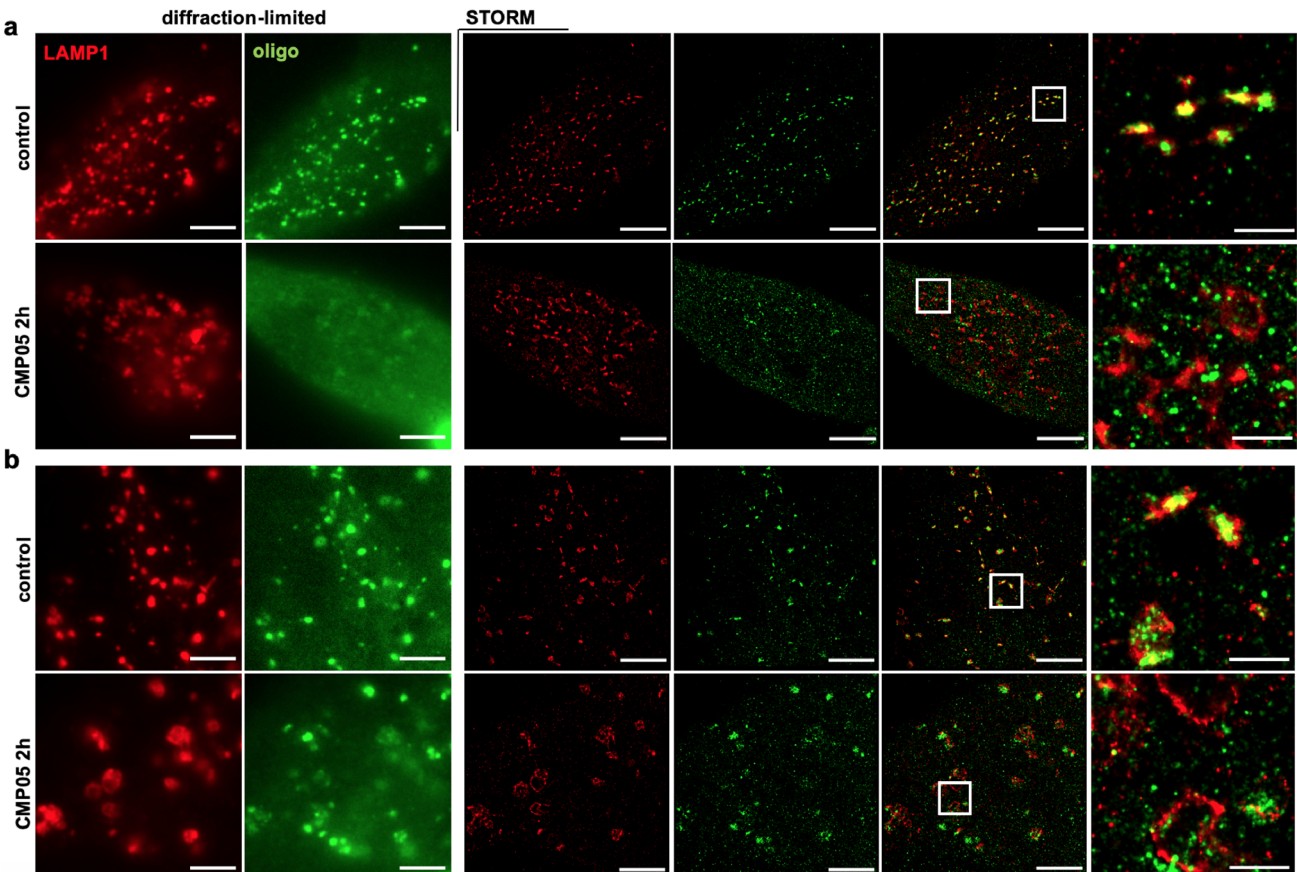

**Fig. 5 STORM imaging of SSO trafficking in CMP05-treated HeLa and HuH7 cells. a** Diffraction-limited and STORM images of lysosomes (red; LAMP1 ICC) and Alexa-Fluor 488-labeled SSO (green) in control HeLa cells and after 2 h treatment with 5 μM CMP05. Scale bars 5 μm, except in the zoomed area 1 μm. **b** Diffraction-limited and STORM images of lysosomes (red; LAMP1 ICC) and Alexa-Fluor 488-labeled SSO (green) in control HuH7 cells and after 2 h treatment with 2.5 μM CMP05. Scale bars 5 μm, except in the zoomed area 1 μm.

cytosolic delivery of ONs in a manner independent of the molecular mechanism of the ON. The desired characteristics of our candidate compounds include efficacious activity in a low-micromolar range and minimal toxicity or other deleterious effects to the recipient cells. This is of course a delicate balance to pursue, as there is a correlation between toxicity and potency for compounds that enhance delivery.

In this functional screen, the novel candidate compound, termed CMP05, was able to increase ON activity in a more efficacious manner than the previously published compound, CMP01, in HeLa and HuH7 cell lines. The rationale behind choosing the specific cell lines for this study is that HeLa is one of the most commonly used cell lines across several fields of cellular biology, and the HuH7 cells are specifically of hepatic origin and are a common cell target for ON therapeutics.

The criteria we chose for compound activity in regards to treatment time and concentration were intentionally strict. The compounds needed to increase functional ON delivery, as measured by quantitation of the functional luciferase splice variants, within minutes to hours. The concentration of compound which should initiate this quick response must also be substantially lower than the concentration of commonly used CADs such as chloroquine, which is used at concentrations between 50 and 100 μM. Although the mCherry-GAL9 assay revealed that many of the screened compounds were capable of inducing endosomal rupture within a 4-h window, CMP05 and CMP05-7 were the only members of our novel class which induced endosomal rupture in less than 2 h at 5 μM and led to the level of ON activity we sought. Other compounds showed modest efficacy and

CMP07 showed efficacy similar to that of CMP01. However, we proceeded with only CMP05 and not CMP07 in our characterization experiments as the different molecular structure of CMP07 suggests a high likelihood that the compounds work through different mechanisms. We additionally proceeded with the CMP05 analog, CMP05-7, due to the efficacy identified in the initial screen, and its structural similarity with CMP05. Importantly, the structures of these compounds do not imply a mechanism in which activity is dependent on an optimal charge ratio with the co-administered ON, as is the case with traditional lipid-based reagents such as lipofectamine 2000 (LF2000). Therefore, higher concentrations of ON could be used during co-treatment with CMP05 and CMP05-7 than could be used with LF2000.

The compound-induced increase in activity was present in both a time and concentration-dependent manner for both compounds, although CMP05-7 was better tolerated by cells for longer treatment durations and at higher concentrations than CMP05, as observed by nuclear morphology. Importantly, acute toxicity was not observed at levels lower than 15 μM. LDH is a marker of plasma membrane perturbation, a common mechanism of compound-mediated toxicity. At 2 h, LDH was not detected in the cell media at increased levels for either CMP05-7 or CMP05 at concentrations up to 15 μM. The results indicate that these compounds could be used in different transfection protocols, for example, CMP05 would better suit a protocol which calls for fast-acting, pulse-exposure-based transfection reagent while CMP05-7 is more applicable as a transfection reagent with a wider treatment window.

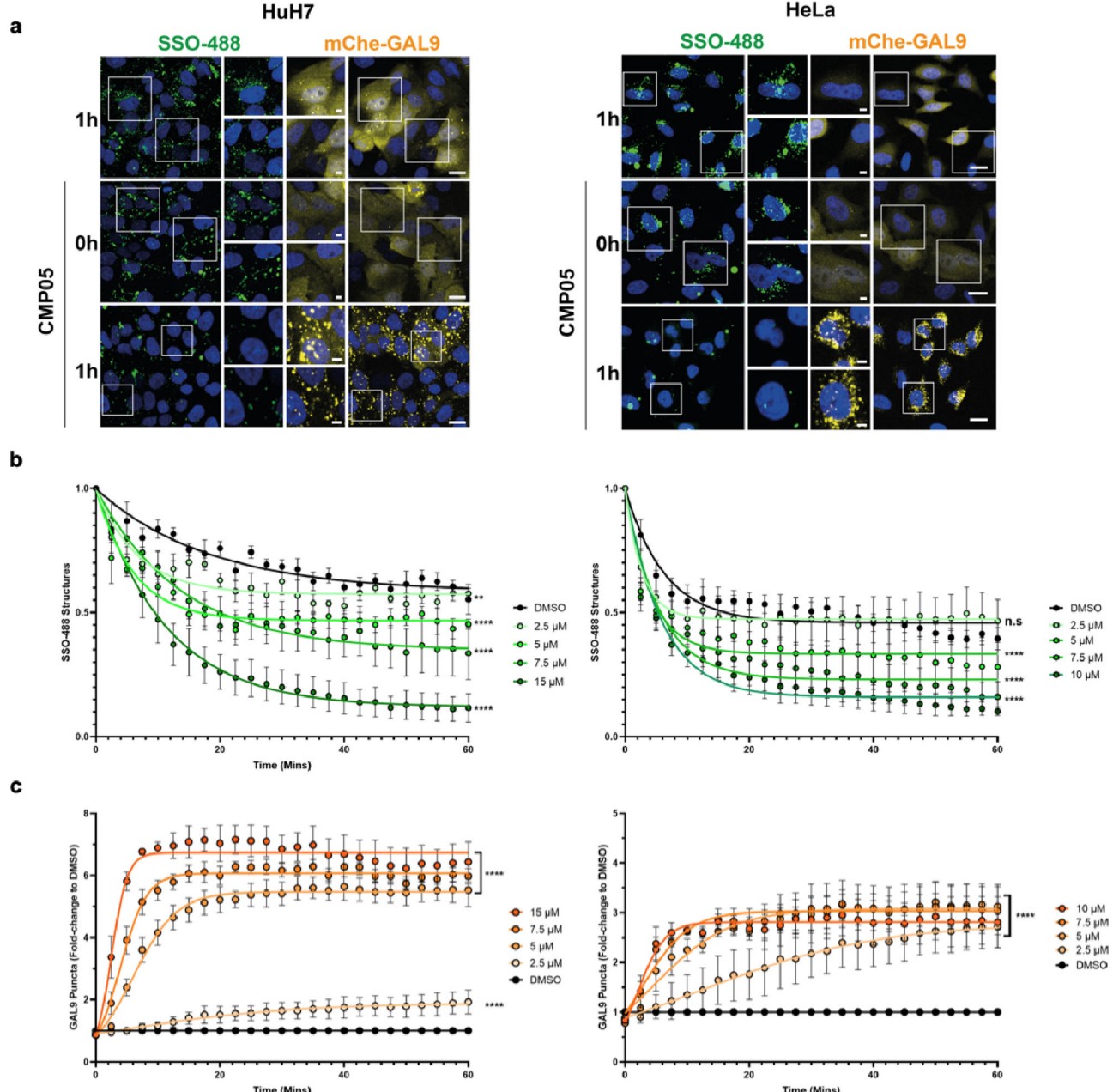

**Fig. 6 Live-cell mCherry-Gal9 assay with SSO-488 in HuH7 and HeLa cells. a** Live-cell confocal microscopy reveals a simultaneous decrease of SSO-containing bodies and Gal9 translocation. HuH7_mCherry-Gal9 and HeLa_mCherry-Gal9 were pre-loaded with SSO-488 (green) for 24 h and then treated with various concentrations of CMP05 and Hoechst to visualize nuclei (blue). Gal9 events appear as yellow puncta. Scale bars = 20 μm (larger panels) and 5 μm (insets). White boxes indicate inset regions. **b** Quantitative analysis of cells treated as in a and imaged every 2.5 min. Analysis reveals a time and dose-dependent decrease in the total number of SSO-488 structures per cell. Data represent mean −/+ SEM from n = 3 independent experiments. **c** Cells treated as in (**b**) and quantitation of mCherry-Gal9 puncta formation. Analysis displays a dose and time-dependent increase of Gal9 structures per cell. Data represent mean −/+ SEM from n = 3 independent experiments. Significance was determined by two-way ANOVA followed by Dunnett's posttest to the DMSO control where ****P < 0.0001 and n.s = not significant.

We hypothesize the mechanism driving CMP05 and CMP05-7 is dependent on endosomal maturation, in a manner by which the compound is able to buffer the lumen of the endosomal compartment by accepting protons at acidic pH[23,35]. Under normal circumstances, the endosomal lumen would decrease in pH, becoming more acidic. In the presence of these compounds, more protons must be pumped into the endosomal lumen in order to achieve the same drop in pH. The resulting accumulation of ions in the lumen may induce osmotic swelling of the endosomal compartment and ultimately membrane rupture or leakage. We

approached this by treating cells with bafilomycin, a potent proton-pump inhibitor, simultaneously with CMP05. This experiment demonstrated that the compound-induced SSO activity is indeed dependent on active proton transport in the endosomes. In addition, by employing STORM to visualize SSO and LAMP1-positive compartments after treatment with CMP05, we confirm that the leakage of SSO into the cytosol occurs in parallel with the physical engorgement of LAMP1-positive late endosomes and lysosomes.

Although our data and microscopy results support a endosomal buffering-related mechanism leading to endosomal escape of

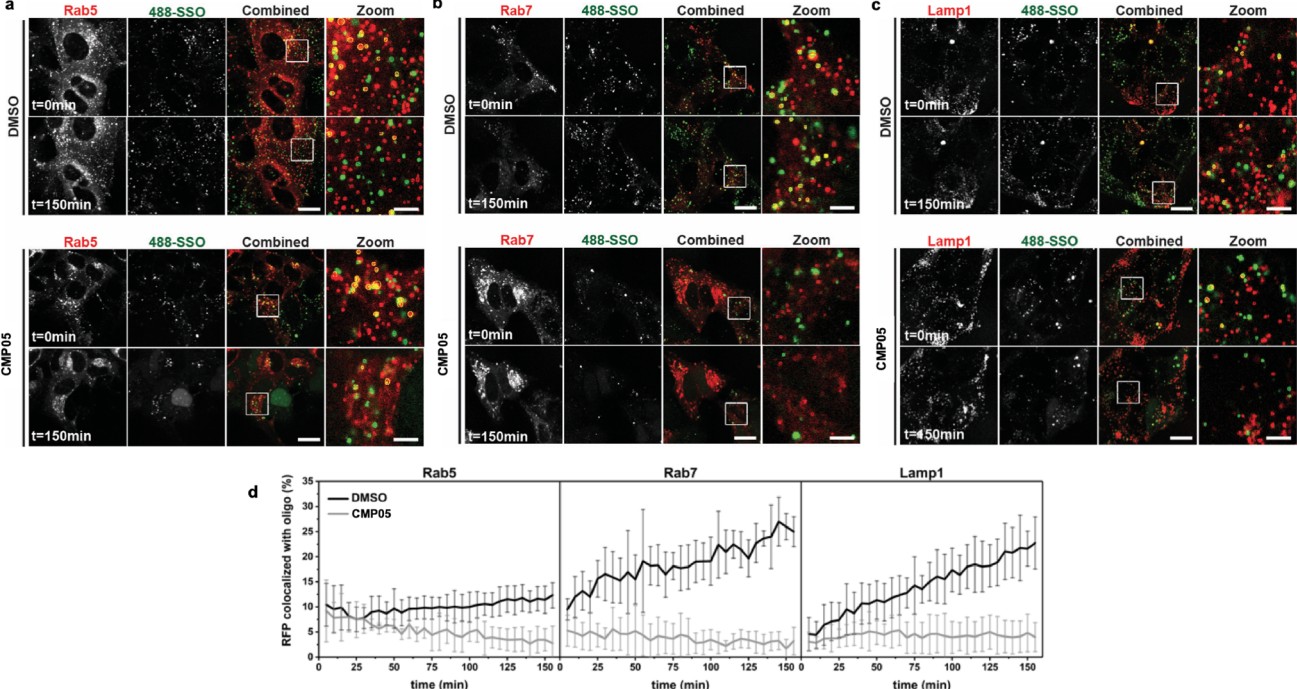

**Fig. 7 Live-cell confocal microscopy images of SSO-endosome colocalization assay. a–c** Live-cell microscopy of HuH7 cells overexpressing mRFP-Rab5 (**a**), mRFP-Rab7 (**b**) or Lamp1-RFP (**c**) after 15 min pulse-incubation with 1 µM SSO and 2.5 µM CMP05. The cells were continuously exposed to 2.5 µM CMP05 and imaged over time. The images were analyzed for colocalization between 488-SSO and (m)RFP by the ComDet plugin to ImageJ and detected particles are marked with a circle (green for 488-SSO, red for (m)RFP and yellow for colocalized particles). Scale bars are 20 µm in the full images and 5 µm in the zoomed images. **d** Quantitation of (m)RFP colocalized with 488-SSO using the ComDet plugin. The data are presented as the percentage of detected (m)RFP-positive vesicles colocalized with SSO-positive vesicles ± SD (CMP05: Rab5 $n = 4$, Rab7 $n = 4$, Lamp1 $n = 5$; DMSO: all cell lines $n = 6$).

the SSOs, we cannot exclude that other mechanisms play a role. For example, these compounds could target a kinase which modulates the morphology of the endosomal bodies and destabilizes the membrane. Also, we cannot rule out the possibility that the compounds alter the rate of vesicle transport or the extent of membrane budding, however, we have not seen any evidence for this in our advanced imaging studies of CMP05. Alternative mechanisms such as these may explain why some of the initially screened compounds induced puncta formation in the mCherry-GAL9 assay without promoting functional SSO delivery, or vice versa.

Next, we sought to correlate the splicing data to the kinetics profiles of the compounds. By using the mCherry-Gal9 assay to characterize the endosomal-rupture response to CMP05 in HeLa and HuH7 cells, we were able to visualize and quantitate the endosomal-rupture dose–response across several concentrations and timepoints. The formation of mCherry-GAL9 puncta occurs in a fashion with strikingly similar kinetics as the loss of SSO puncta across all concentrations of compounds used. The correlation implies that the loss of punctate SSO localization is due to SSO escaping from the endosomes into the cytosol due to the membrane rupture which is responsible for the recruitment of mCherry-GAL9. The loss of SSO puncta, induction of GAL9 recruitment, and the functional splice-switching data all correlate closely. The steep dose–response curves at the higher concentrations and the lack of compound activity at the lower concentrations, even at longer timepoints, indicate a threshold effect for SSO release rather than a gradual response.

To conclude the characterization of CMP05, we utilized a colocalization-based live-cell tracking assay in which SSOs can be colocalized with early endosomes, late endosomes, and lysosomes separately to determine at which point in the endocytic pathway the ON escapes. In this assay, SSO-containing bodies can be

quantitatively colocalized with each endosomal compartment. Typically, as SSO is internalized and trafficked within the cell, the number of SSO-containing early endosomes would be expected to remain fairly constant or decrease slightly as oligonucleotide is trafficked through the early endosomes and into other endosomal bodies. SSO will then accumulate within the late endosomes and lysosomes, where it becomes sequestered. We demonstrate this pattern by administering SSO with DMSO, a vehicle control, and subsequently observe a steady increase in the colocalization of RFP-Rab7 and RFP-LAMP1 to the SSO. However, in the presence of endosomolytic compound, the RFP-Rab7 and RFP-LAMP1 colocalization do not occur, indicating the SSO is no longer becoming sequestered in the late compartments. Importantly, we do note a slight decrease in the overall number of RFP-Rab5-positive bodies after treatment with CMP05, indicating a decrease in the total number of early endosomes during the 2 h in which images were acquired. The reduction of RFP-Rab5 positive bodies is in agreement with findings from others and support that administration of endosomolytic compounds can alter the kinetics of endocytosis and thus the formation of early endosomes, and that this effect is reversible[34].

Ongoing work is focused on finding further applications for this class of compounds and continuing to screen for functional analogs. Ideally, a compound would be able to encourage intracellular delivery without needing a media-change step, to simplify the transfection. We are also exploring the compatibility of other macromolecular cargos in vitro, including other oligonucleotide chemistries and potentially proteins. Thirdly, work is ongoing to find a suitable in vivo application. Using small molecules to enhance the delivery of other therapeutics in vivo has been challenging for various reasons. For one, the small molecules may be easily cleared by the body so that their in vivo efficacious window is much smaller than the corresponding activity window

in vitro. Others have addressed this by administration of ASOs and compound in a clinically irrelevant manner such as intraperitoneal injection. Ongoing work is focusing on finding a clinically relevant application for our compounds.

In conclusion, we have identified and conducted a functional characterization of two novel endosomolytic compounds with favorable properties to induce endosomal escape of ONs. These compounds are able to efficiently induce an increase in SSO activity at concentrations below 10 µM. Furthermore, we have utilized several state-of-the-art methodologies for the characterization and optimization of these compounds. This includes live-cell imaging techniques capable of visualizing the endosomal-rupture events as they occur in real time. The rupture was then examined on the super-resolution scale and we were able to visualize the endosomal engorgement which occurs simultaneously with ON leakage. Finally, we were able to uncover the altered trafficking patterns of SSOs in the endolysosomal system. The employment of these techniques allows us to investigate the altered-ON trafficking patterns with a high level of certitude. Collectively, these findings open new opportunities to endosomolytic compound design and provide a platform for future screens to investigate ON delivery strategies.

## Methods

**Splice-switching ONs (SSOs).** Luc705-targeting ONss (705-SSO, A488-SSO, and A568-SSO) were purchased from Integrated DNA Technologies and synthesized with 2′-O-methylated (2′-OMe)-modified bases and phosphorothioate (PS)-saturated backbone linkages. For the non-fluorescent ONs, HPLC purification and Na+ salt exchange steps were included. SSOs were shipped and stored in IDTE buffer at a pH of 8.0 at 100 µM. The sequences of the ONs, including those with fluorescent modifications, are listed in Table 1.

**Small-molecule selection.** The characterized compounds came from the AstraZeneca internal small-molecule collection with the addition of one previously published compound, UNC2383 (herein referred to as CMP01), known to enhance endosomal escape[18].

**Small-molecule synthesis.** See Supplementary Methods.

**Cells and culture methods.** HeLa_Luc705 and HuH7_Luc705 cells, as described by Rocha et al. were used to screen and characterize compounds[26,27]. Here, a plasmid carrying the luciferase coding sequence interrupted by an insertion of intron 2 from β-globin pre-mRNA carrying a cryptic splice site is stably transfected into cells. Unless the aberrant splice site is masked by SSO, the pre-mRNA of luciferase will be incorrectly processed. Thus, by using these cells, various SMCs can be evaluated by co-treating with SSOs and measuring luciferase activity.

Cells used for imaging in the GAL9 assays included HeLa_mCherry-Gal9 and HuH7_mCherry-Gal9. Cells used in the endosomal colocalization assay included Huh7 cells (WT and in-house established cell lines overexpressing mRFP-RAB5, mRFP-RAB7, or LAMP1-RFP). All stable cell lines used were maintained and cultivated in growth media which comprised of: Dulbecco's modified Eagle's medium (DMEM) with high glucose with GlutaMAX (Gibco, cat. no. 31966-021) and 10% fetal bovine serum (Gibco, cat. no. 10270-106). All cell lines except for the lines overexpressing mRFP-RAB5, mRFP-RAB7 or LAMP1-RFP were also grown with 1% penicillin-streptomycin (Gibco, cat. no. 15140-122). All cell lines were grown at 37 °C, 5% CO$_2$ in 95% humidity in T75 TC flask (Sarstedt, 83.3911.002). The cells were detached with trypsin 0.25% EDTA (Gibco, cat. no. 25200056) for 10 min and passaged twice a week. HuH7 and HeLa_mCherry-Gal9 cell lines growth media was supplemented with 1 µg/ml puromycin (Gibco, cat. no. A11138-03) to maintain reporter expression. All cells were routinely tested and mycoplasma negative.

**Splice-switching activity assays.** HeLa_Luc705 and HuH7_Luc705 cells were seeded in 96-well plates at 10,000 cells per well in growth media with 1 µM or 100 nM Luc705-SSO. SMCs were diluted directly into prepared growth media. After 24 h to allow adherence and SSO internalization, 10 µl of diluted SMC was added into each well at indicated concentrations. For positive control, SSOs were complexed with Lipofectamine 2000 for 30 min at RT and cells treated with a final concentration of 200 nM. ON concentrations above 200 nM would require LF2000 to be used in excess of the cells toxicity threshold. After treatment, media was removed from the wells and replaced with fresh growth media for 4 h to allow translation of luciferase. The media was then removed again and cells lysed with 0.1% Triton X-100 (Sigma Aldrich, cat no. X-100) in 1× PBS (Gibco, cat no. 10010023).

For the detection of luciferase activity, 30 µL of cell lysate was transferred to white-walled 96-well plates (Sigma Aldrich, cat no. CLS3922). The luciferase intensity in each well was immediately measured ($n = 3$, $N = 3$) using a GloMax® 96 Microplate Luminometer machine (Promega) following auto-injection of 25 µL Luciferin substrate per well as per the Promega Firefly Luciferase Assay System (Thermo Scientific, cat no. 16174). The exposure time of photon measurements was set to 10 s, delayed 2 s after injection.

**Lactate dehydrogenase (LDH) cell toxicity assay.** Conditioned cell media was retained after 2-h co-treatment with SSOs and the compounds. In parallel, cell-death positive LDH controls were collected by lysing cells with 0.1% Triton X-100 in PBS. LDH activity was measured ($n = 3$, $N = 3$) with the CyQUANT™ LDH Cytotoxicity Assay (ThermoFisher Scientific, cat no. C20300) according to the manufacturer's instructions. Cell toxicity was reported as the percentage of released LDH against the cell-death positive control.

**RT-PCR.** Quantification of the percentage corrected luciferase mRNA was accomplished utilizing a previously validated RT-PCR protocol[36]. Total RNA was extracted from HeLa_Luc705 and HuH7_Luc705 cells using TRI Reagent (Sigma Aldrich, cat no. T9424) according to the manufacturer's instructions. For RT-PCR reactions using ONE STEP RT-PCR kit (QIAGEN, cat no. 210210), three nanograms of isolated RNA were utilized. The total reaction volume was 20 µL and the primer sequences used were: Fwd-5′-TTGATATGTGGATTTCGAGTCGTC-3′; Rev-5′-TGTCAATCAGAGTGCTTTTGGCG-3′. The program for the RT-PCR was as follows: 55 °C, 35 min, then 95 °C, 15 min, for the reverse transcription step, directly followed by the PCR (94 °C, 30 s, then 55 °C, 30 s, then 72 °C, 30 s) for 30 cycles and the final extension 72 °C, 10 min. The PCR products were analyzed using a 1% agarose gel in 0.5× TAE buffer and visualized by SYBR Gold (ThermoFisher, cat no. S11494) staining.

Versadoc imaging system with a cooled CCD camera (BioRad, Hercules, CA, USA) was used to acquire representative images for the analysis of gels ($n = 1$, $N = 2$). Band intensities were analyzed with the Quantity One software (BioRad). The percentage of correction was calculated using this equation: (Band intensity of corrected RNA/(Band intensity of corrected RNA + Band intensity of uncorrected RNA) * 100%.

**Immunocytochemistry.** For STORM, HeLa_Luc705 and HuH7_Luc705 cells were seeded on eight-well glass-bottom µslides (Ibidi, cat no. 80807) at a density of 20,000 cells/cm$^2$ and 40,000 cells/cm$^2$, respectively. SSO-Alexa-Fluor 488/SSO-Alexa-Fluor 568 was added to the culture medium at 1 µM concentration upon seeding. Cells were let to attach for 24 h in the cell culture incubator, after which they were treated for 2 h with CMP05 (5 µM for HeLaLuc705 cells and 2.5 µM for HuH7_Luc705 cells), washed once with PBS and fixed with 4% electron microscopy grade paraformaldehyde (Fisher Scientific, cat no. 50-259-99) and 0.2% electron microscopy grade glutaraldehyde (Merck, cat no. 104239) in PBS at RT for 15 min. Both PBS and fixation solutions were warmed up to +37 °C and the pipetting was conducted swiftly to avoid sample drying. Following fixation, cells were permeabilized with 0.05% Triton X-100 in PBS for 5 min at RT. Blocking was performed with 3% bovine serum albumin (Fisher Scientific, cat no. 11423164) in PBS for 2 h at RT. Primary antibody staining the lysosomal marker LAMP1 (D2D11, Rabbit mAb, Cell Signaling Technology, cat no. 9091S), diluted 1:200 in the blocking solution, was incubated at 4 °C overnight. Secondary staining with donkey anti-rabbit IgG Alexa-Fluor 647 (Invitrogen, cat no. A-31573), diluted 1:1000 in the blocking solution, was performed at RT for 90 min. Finally, to fix the antibodies a post-fixation of the samples was conducted with 2% PFA in PBS for 10 min at RT. STORM experiments were performed in three separate wells for each

**Table 1 Nomenclature, sequence, chemical modifications, and size of SSOs.**

| Name | Sequence | Size (nt) |
|---|---|---|
| Luc705-SSO | 5′-mC*mC*mU*mC*mU*mU*mA*mC*mC*mU*mC*mA*mG*mU*mU*mA*mC*mA-3′ | 18 |
| A488-SSO | 5′-Alexa-488-mC*mC*mU*mC*mU*mU*mA*mC*mC*mU*mC*mA*mG*mU*mU*mA*mC*mA-3′ | 18 |
| A568-SSO | 5′-Alexa568-mC*mC*mU*mC*mU*mU*mA*mC*mC*mU*mC*mA*mG*mU*mU*mA*mC*mA-3′ | 18 |

*PS linkage, *m* 2-OMe nucleotide.

condition with multiple imaging fields per well for total replicates of: HeLa control, $n = 7$; HeLa treated, $n = 6$; HuH7 control, $n = 7$; HuH7 treated, $n = 7$.

**Stochastic optical reconstruction microscopy (STORM)**. Immediately before imaging, the sample was soaked in imaging buffer with the following composition: Tris buffer (160 mM Tris, 40 mM NaCl, pH adjusted to 8.0), 10 wt-% glucose, 0.5 mg/mL glucose oxidase from *Aspergillus niger* (Sigma Aldrich, cat no. G7141), 47 µg/mL catalase from bovine liver (Sigma Aldrich, cat no. C1345), and 10 mM MEA (cysteamine, pH adjusted to 8.0). The plate was sealed with parafilm to minimize oxygen entry. Stochastic optical reconstruction microscopy was conducted with Nikon Ti Eclipse inverted microscope (Nikon, Tokyo Japan), housing cube filters (excitation: Chroma ZET405/488/561/640x, emission: Chroma ZET405/488/561/640 m) and TIRF dichroic ZET405/488/561/640bs, equipped with Cairn laser module (Cairn Research, Kent, UK) with 200 mW 488, 150 mW 561, and 140 mW 642 lasers used in this study. CFI SR Apo TIRF ×100 oil objective (N.A. 1.49) was used, in combination with ×1.5 Optovar lens, resulting in a final magnification of ×150. The camera (Andor iXON Ultra 888 EMCCD, Oxford Instruments, Belfast, UK) had a pixel size of 13 µm, giving a final pixel size of 87 nm with the ×150 magnification. The image acquisition was controlled with MetaMorph and Micro-Manager software. For each STORM acquisition, a 256 × 256-pixel region of interest (ROI) was imaged. A widefield diffraction-limited image was taken from each ROI for reference before starting the STORM acquisition, for which the laser power was increased to 100%. The acquisition was only started when the photoswitching of the fluorophores reached an optimal level. Around 30,000 frames, with an exposure time of 30 msec/frame, were recorded for each image. Electron multiplying gain of 100 and 200 was used for 647/561 and 488 channels, respectively. In two-color STORM imaging, the channels were recorded sequentially, and the higher wavelength was recorded first to minimize the damage caused to the fluorophore in the other channel.

The images were reconstructed with the ThunderSTORM plugin in Fiji[37] followed by drift correction using the ThunderSTORM cross-correlation algorithm. Moreover, images were filtered based on the sigma values (standard deviation of the Gaussian fit over the point spread function of each blink) to remove low-quality signals (e.g., noise and partially overlapping fluorophore signals). For the lysosome size quantification, additional intensity-based filtering was conducted to further remove background and noise and thus enable higher-quality feature detection. For a more detailed description of the reconstruction and post-processing parameters, see Supplementary Methods. All the images were visualized with the Normalized Gaussian visualization method using a magnification of 10 and an image-specific uncertainty constant.

**STORM image analysis**. STORM images were segmented using the software package ilastik (version 1.3.3post2, EMBL)[38]. A pixel classification pipeline was created and foreground and background pixels annotated on the first replicate of each treatment. The trained classifier was used to generate a pixel prediction map for each image. This map was incorporated into a second object classification pipeline in ilastik. A threshold was applied to the prediction map in "hysteresis" mode, with a core threshold of 0.95 and a final threshold of 0.85, and a size filter of $1 \leq$ pixel count $\leq 1{,}000{,}000$. Background "small" lysosome fragments and "large" lysosome objects were hand-annotated on a selection of images. The trained classifier was used to produce segmentation masks for both objects (individual objects) and classes ("Large" or "Small"). The geometry of the segmented objects was assessed using the software package CellProfiler (version 3.1.9, Broad Institute)[39]. This pipeline filters out objects touching the image border, generates quality control images of the object-level segmentation and makes multiple measurements of the shape of the binary objects. The validity of the segmentation was assessed qualitatively by visually inspecting the quality control images. Data were analyzed with the R programming language (version 3.6.3)[40] using the packages: tidyverse[41], here[42], and broom[43]. Intact lysosomes were identified by the "Large" class of objects, and the diameter of the objects compared using Welch's $t$ test for unequal variances. For full details of the analysis see the R Notebook provided as Supplementary Information.

**STORM reconstruction and post-processing parameters**. The following parameters were chosen for the STORM image reconstruction in the ThunderSTORM software:

Image Filtering

- Filter: Difference-of-Gaussians filter (Sigma1 = 1.0 px, Sigma2 = 1.6 px)

Approximate localization of molecules

- Method: Local maximum
- Peak intensity threshold: std(Wave.F1)
- Connectivity: 8-neighborhood

Sub-pixel localization of molecules

- Method: PSF: Integrated Gaussian
- Fitting radius (px): 3
- Fitting method: Weighted least squares

- Initial sigma (px): 1.6
- Multi-emitter fitting analysis: enabled
- Maximum of molecules per fitting region: 3
- Model selection threshold ($P$ value): 1.0E-6

The intensity range (photons) is not limited.

For the final visualization, a Normalized Gaussian method was used, with a magnification of 10 and an uncertainty value calculated image-specifically. In two-color images, the visualizations of each channel were merged and aligned manually in Fiji. Regarding image post-processing, in addition to drift correction with a cross-correlation algorithm, sigma-based filtering was conducted to remove noise/background in the lower end and signal from partially overlapping fluorophores in the upper end. The lower and upper limits for sigma filtering were calculated channel-specifically using the full width at half maximum (FWHM) formula for resolution (FWHM = $\lambda/2NA$, where $\lambda$ was the wavelength (488/562/647) and NA = 1.49). This gave the upper limit for the sigma filtering. The lower limit was set to one standard deviation of the fitted Gaussian curve (FWHM/2.3). The upper and lower sigma filtering values for different channels were the following: 647 channel [90, 215], 562 channel [80, 190], and 488 channel [70, 165]. In addition to sigma filtering, for the quantification of lysosome sizes, additional intensity-based filtering was conducted to further remove background signal/noise and thus improve the robustness of the quantification method. For this filtering, a fixed lower limit of 1500 was chosen based on visual evaluation and used for all the images.

**mCherry-GAL9-recruitment assay**. For experiments examining endosomal damage and mCherry-GAL9 recruitment, HeLa or HuH7_mCherry-Gal9 expressing cells were seeded into 384-well CellCarrier Ultra plates (PerkinElmer, cat no. 6007558) at 3000 or 3500 cells/well, respectively, 16 h before experimental usage.

Dose–response curves of indicated compounds were dispensed by utilizing an Echo 655T acoustic dispenser (Labcyte) into source plates (Greiner, cat no. 781280) containing growth media. At experimental start points, media containing appropriate compounds and doses was transferred to plates using a liquid handling robot (Agilent Bravo). At assay endpoints, cells were washed 2× PBS at RT and fixed in 4% PFA (VWR, cat no. 9713.1000) for 15 min at RT. Cells were washed a further 3× PBS before the addition of PBS + 1 µg/ml Hoechst 33342 (ThermoFisher Scientific, cat no. H21492) for a minimum of 1 h before imaging. For time-lapse experiments, cells were imaged within a humidified environmental chamber that was maintained at 37 °C and supplemented with 5% $CO_2$."

Plates were imaged using a spinning-disk confocal microscope (Yokogawa: CV7000) with a ×20 objective (NA 0.75). Images were processed utilizing Columbus image-analysis software (PerkinElmer: v2.9.0) to identify and quantify cells and mCherry-GAL9 structures. The resulting data were processed and normalized in Spotfire (Tibco: v10.3) and plotted in Prism (Graphpad: v8.0.1). Image panels were assembled using the FigureJ plugin for FIJI[44]. All Gal9 experiments were performed three times ($N = 3$) with two separate image fields in every well ($n = 2$).

**SSO-endosome colocalization live-cell assay**. HuH7 cells (wt, mRFP-RAB5, mRFP-RAB7, or LAMP1-RFP) were plated 1 day before the experiment in CELLview™ glass-bottom quartering cell culture dishes (Greiner Bio-One, cat no. 391-0252), 45,000 cells in 250 µl/well. For studying intracellular tracking of SSO, cells were washed 1× with cell culture medium and pulse-incubated with 1 µM SSO and either 2.5 µM CMP05 (final DMSO concentration 0.25%) or corresponding concentration of DMSO in complete medium for 15 min. The cells were thereafter washed 1× with cell culture medium and imaged immediately in complete medium with continuous exposure to 2.5 µM CMP05 (or corresponding concentration of DMSO).

Confocal images were acquired on a Nikon C2 + confocal microscope equipped with a C2-DUVB GaAsP Detector Unit and using an oil-immersion ×60 1.4 Nikon APO objective (Nikon Instruments, Amsterdam, The Netherlands). Pinhole size was set to 90 µm (corresponding to 3.3 AU) to detect particles throughout the volume of the cell, as well as to increase the signal and hence particle detection. The sample was excited and detected with appropriate excitation laser lines and emission filters and the A488-SSO and RFP fluorophores were imaged sequentially. The samples were imaged every 5 min with a field of view of 187 × 187 µm, containing in the range of at least 15 cells/frame.

The images were analyzed in the ImageJ software (NIH, Bethesda, Maryland, USA) with the spot colocalization ComDet plugin[45]. Approximate particle size was set to four pixels, the threshold set to 15% for both channels, larger particles were segmented and maximum distance between colocalized spots was set to four pixels. The data was analyzed and visualized in Origin software (OriginLab, Northampton, MA, USA). The number of detected particles was normalized to the number of particles detected in the first frame ($t = 0$ min). The experiment was repeated on six separate occasions ($N = 6$). Samples containing cells displaying CMP05-originating toxicity within the field of view were removed from the analysis and analysis was hence performed on $n = 3–6$ (CMP05: WT $n = 3$, Rab5 $n = 4$, Rab7 $n = 4$, Lamp1 $n = 5$; DMSO: WT, Rab5, Rab7 and Lamp1 $n = 6$). The results are displayed as mean ± SD.

**Statistics and reproducibility**. For functional compound analysis in the splice-switching assay, results are presented as mean ± SD. The number of experimental replicates is denoted in each figure legend, with a minimum number of three biological replicates ($n = 3$) and the experiment being run a minimum of three times ($N = 3$). Statistical significance (*$P < 0.05$, **$P < 0.01$, ***$P < 0.001$, and ****$P < 0.0001$) between compounds with multiple concentrations was calculated with two-way ANOVA with no matching and Dunnett's posttest against a control condition, denoted in the figure legends. In STORM analysis, the diameter of the objects was compared using Welch's $t$ test for unequal variances. For quantitative analysis in the Gal9 endosomal-rupture assay, results are presented as mean ± SEM.

**Reporting summary**. Further information on research design is available in the Nature Research Reporting Summary linked to this article.

## Data availability
The data that support the findings of this study (data DOI: 10.5281/zenodo.5833599) are openly available for download under Creative Commons Attribution 4.0 International license under the repository name "Novel endosomolytic compounds enable highly potent delivery of antisense oligonucleotides" at https://doi.org/10.5281/zenodo.5833599[46].

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

## Acknowledgements
M.O. was funded by grants from the Jane and Aatos Erkko Foundation, Otto A. Malm Foundation, The Paulo Foundation, and the Oskar Huttunen Foundation. O.S. was supported by a PhD grant from the Egyptian Ministry of Higher Education. M.N.H. acknowledges support from the Swiss National Science Foundation (P300PA_171540). M.M.S. and S.G.H. acknowledge a Wellcome Trust Senior Investigator Award (098411/Z/12/Z). Parts of this study were performed at the LCI facility/Nikon Center of Excellence, Karolinska Institutet, supported by grants from the Knut and Alice Wallenberg Foundation, Swedish Research Council, KI infrastructure, Centre for Innovative Medicine, and Jonasson center at the Royal Institute of Technology. This work has been supported by grants from the Swedish Foundation of Strategic Research (SSF) in the Industrial Research Centre, FoRmulaEx – Nucleotide Functional Drug Delivery (IRC15-0065), and the AstraZeneca R&D PostDoc program.

## Author contributions
J.P.B., M.O., M.J.M., E.W., A.G., D.G., O.G., O.S. and J.R. performed experiments. J.P.B. and M.O. carried out STORM imaging. J.P.B. and M.J.M. carried out Gal9 assays. E.W. and A.G. performed SSO-endosome colocalization assays. O.S. performed PCR experiments. M.O. and S.G.H. carried out STORM analyses. Study design and critical discussions were carried out among T.L., M.N.H., A.D., O.E., P.E.S., S.A., E.K.E., A.C. and S.E.A. S.A., C.I.E.S., M.M.S., E.K.E., A.C. and S.E.A. provided funding and project direction. The manuscript was written by J.P.B. and S.E.A., with input from all authors.

## Funding

## Competing interests
M.J.M., A.G., A.D., O.E., P.E.S., S.A. and A.C. are employees of AstraZeneca plc. The remaining authors declare no competing interests.
