## [Peer Review File · Communications Biology]

Reviewers' comments:

Reviewer #1 (Remarks to the Author):

The paper by Bost et al identifies novel small molecules that increase endosomal escape. State of the art microcopy, novel galectin assay was used to show improved subcellular delivery of SSO's. This is an interesting study. I have minor comments that need to be addressed.

1. Do these small molecules enhance delivery of other compounds like siRNA or mRNA
2. The small molecule screens have been done before but none show an improved in-vivo delivery. Please comment
3. The authors should show improved delivery in hard to transfect cells than HeLa or Huh-7.

Reviewer #2 (Remarks to the Author):

The authors present characterization of two new related endosomolytic small molecule compounds to assist with delivery of splice-switching oligonucleotides. The in vitro assays including real-time assessment of endosomal escape and STORM are compelling, but the authors are asked to consider the following suggestions.

Major Comments:

1. Activity / Toxicity Comparisons. None of the CMP05 derivatives outperformed the parent compound 05, and CMP05-7 also underperforms compared to the existing CMP01, so the justification for including these and moving forward with CMP05-7 and not also including CMP07 could be stronger. CMP05-7 is relatively less toxic than CMP05, so perhaps use of CMP05-7 at higher concentrations than tested (in the 20 μM + range) might be more favorable. The authors are recommended to include a higher-resolution dose curve where both cytotoxicity and functional activity measured under the same conditions for CMP05, CMP05-7, and CMP01 (and ideally CMP07) in order to give the clearest possible picture of the activity/toxicity tradeoffs for the compounds. This is especially warranted since there is a significant dropoff in activity at 10 μM and 1 μM vs 5 μM for the lead compound 05 in the existing Figure 1b data. Related to the toxicity analyses, the toxicity data in Fig. 2b taken immediately after 2 hours of treatment show toxicity for CMP05 in both HuH7 and HeLa cells, with the therapeutic window in the latter being quite narrow (\sim 2-4 μM). It would be helpful to assess toxicity at later timepoints (24 hrs would be standard), following at least 2 hours of treatment with CMP05/CMP05-07. Additionally, it is unclear why the cell viability is consistently no better than \sim 80% even at extremely low doses of the small molecule drugs in Fig. 2b.
2. Proton Sponge. The claims regarding proton sponge as the mechanism for disruption are a bit circumstantial. Others have shown that active membrane disruption reagents (and not just proton sponge systems) can cause endosome enlargement. Some thoughts that could be considered are a pH-dependent hemolysis or synthetic vesicle disruption assay to more directly determine that the molecules do not have any active membrane disruption activity. It seems unlikely that small molecules would but may be necessary to solidify the conclusion. Though it doesn't separate out membrane active vs proton sponge mechanisms of endosome disruption Bafilomycin A can be an additional nice tool for proving dependency of molecule activity on the endosomal acidifying proton pump (<https://doi.org/10.1038/s41467-019-12906-y>).

Minor Comments:

1. Greater elaboration on the intended application of these new compounds should be discussed. Is this intended to primarily serve as an in vitro transfection enhancer? If so, it would strengthen the manuscript to consider demonstrating with other types of biological cargos. If therapeutic potential is the goal, it would be desirable to demonstrate at least in vitro splice switching of a disease-relevant target vs solely a luc reporter. CMP01 has been previously demonstrated in an animal model for example (reference 18).

2. The authors should consider citing the first paper where Galectin recruitment imaging was utilized for library screening for endosome escape. <https://doi.org/10.1021/acsnano.8b05482>
3. In Figure 1b-c, please clarify the statements that "data comes from n=3 experiments" – does this mean that the experiment was repeated three times, or that a single experiment was conducted with n=3 biological replicates?
4. Statistically significant differences by two-way ANOVA should be shown in Figure 1c. If none of the differences is significant, then this should be indicated in the caption.
5. For line 252/Figure 4, please explain why HeLa and HuH7 cells were treated with different concentrations of CMP05, and notably why HeLa cells received a higher dose when they appear to be the more susceptible cell line to toxicity from CMP05.
6. Please add information about temperature/environmental control for timelapse experiments (e.g. Figure 6) to the methods section. If these experiments were conducted at room temperature, please address whether this could impact the kinetics of the observed phenomena.

To the Reviewers,

December 2nd, 2021

We are now ready to resubmit our article, “**Novel endosomolytic compounds for highly potent delivery of antisense oligonucleotides**” with manuscript ID **COMMSBIO-21-1546A** by Jeremy Bost et al. The review comments we received were in many cases warranted and have resulted in a manuscript of superior quality than before the review process. We extend our deepest appreciation to the reviewers.

We have followed the reviewer's suggestions in the majority of instances, and this includes a higher resolution dose curve, reformatting several figures, and changing the introduction, methods, results and the discussion sections where necessary. There were only a few comments we disagree with, and in these cases, we have answered with an argument justifying our belief. We have tried to address all of the raised concerns below in red.

Reviewer #1 (Remarks to the Author):

The paper by Bost et al identifies novel small molecules that increase endosomal escape. State of the art microcopy, novel galectin assay was used to show improved subcellular delivery of SSO's. This is an interesting study. I have minor comments that need to be addressed.

1. Do these small molecules enhance delivery of other compounds like siRNA or mRNA

Outside of this manuscript, we have been able to deliver various other cargoes with these compounds. These include LNA oligos, mRNA-CPP complexes, and sgRNA-Cas9 RNPs. However, the compound does not deliver naked mRNA, likely due to the instability of the mRNA molecule in cell media containing serum. We believe that these experiments lie outside the scope of our main manuscript, as here we have focused methodologically on the transection of oligos. Other cargoes such as proteins and alternatively modified nucleic acids will have different half-lives within endosomal compartments, and therefore likely require cargo-dependent characterization experiments, which we believe makes these experiments suitable for another manuscript.

2. The small molecule screens have been done before but none show an improved in-vivo delivery. Please comment

Using small molecules to enhance the delivery of other therapeutics has been challenging in vivo for a few reasons (that we know of). For one, the small molecules may be easily/quickly cleared by the body, meaning that their efficacious *in vivo* window is much smaller than the corresponding activity window *in vitro*. Others have addressed this with IP administration of oligo and compound, however this is not a clinically relevant approach.

Additionally, in local delivery, there seems to be a very tight window in which the compound enters cells at adequate levels for activity but without inducing an inflammatory response. As we are still working to find a suitable *in vivo* application of our compound, we have intentionally excluded *in vivo* data from this manuscript but we would anticipate that it could work in a local setting. To this end, we believe this work is better suited in a separate manuscript.

To address the reviewer's concern here, we have added text into the manuscript's discussion discussing *in vivo* applications.

3. The authors should show improved delivery in hard to transfect cells than HeLa or Huh-7.

We fully agree with the reviewer. To this end, we have conducted experiments to deliver SSO into both Neuro2A and U2-OS cells expressing the 705-reporter construct. We have added these new results into a new supplementary figure 4. Similar to HeLa cells, the compounds had a significant effect on the splice switching activity.

Postal address

Karolinska Institutet, Dept. of Laboratory Medicine,
Clinical Research Center (CRC), Novum
Hälsövägen 7
SE-141 52 Huddinge, Sweden
Org. number 202100 2973

Visiting address

Clinical Research Center (CRC)
CRC-Novum Science Park
5th floor, Blickagången 6,
Flemingsberg - Huddinge

Contact

Direct +46 (0)8 585 838 73
Lab +46 (0)8 585 836 53
Fax +46 (0)8 585 836 50
E-mail jeremy.bost@ki.se

Reviewer #2 (Remarks to the Author):

The authors present characterization of two new related endosomolytic small molecule compounds to assist with delivery of splice-switching oligonucleotides. The in vitro assays including real-time assessment of endosomal escape and STORM are compelling, but the authors are asked to consider the following suggestions.

Major Comments:

1. Activity / Toxicity Comparisons. None of the CMP05 derivatives outperformed the parent compound 05, and CMP05-7 also underperforms compared to the existing CMP01, so the justification for including these and moving forward with CMP05-7 and not also including CMP07 could be stronger. CMP05-7 is relatively less toxic than CMP05, so perhaps use of CMP05-7 at higher concentrations than tested (in the 20 μM + range) might be more favorable.

We thank the reviewer for raising this. Our justification for proceeding with CMP05 was of course that it is the most efficacious compound we identified in this project. After re-reading our manuscript, we agree that there is not enough justification for why we chose to proceed with CMP05-7 and not CMP07. We only chose one compound from the first screen of 7 compounds because these 7 compounds have very different structures, and therefore a high chance that they work via different mechanisms (for example, CMP07 may work by active membrane disruption, or through protein interaction, or something else entirely). As figure 1 shows, the structures between CMP05 and CMP05-7 are highly similar, and therefore we can confidently hypothesize that they work through the same mechanism. We have added text to the results and discussion sections to clarify why we made this choice.

The authors are recommended to include a higher-resolution dose curve where both cytotoxicity and functional activity measured under the same conditions for CMP05, CMP05-7, and CMP01 (and ideally CMP07) in order to give the clearest possible picture of the activity/toxicity tradeoffs for the compounds. This is especially warranted since there is a significant dropoff in activity at 10 μM and 1 μM vs 5 μM for the lead compound 05 in the existing Figure 1b data.

We appreciate the reviewer's input here, and fully agree that it is necessary to include a higher-resolution curve for both SSO efficacy. This has been added in figure 3, demonstrating a higher resolution dose response. We believe that these experiments, taken together with the survival curves generated in figure 2 provide ample insight into the tox/activity trade-offs.

Related to the toxicity analyses, the toxicity data in Fig. 2b taken immediately after 2 hours of treatment show toxicity for CMP05 in both HuH7 and HeLa cells, with the therapeutic window in the latter being quite narrow (~2-4 μM). It would be helpful to assess toxicity at later timepoints (24 hrs would be standard), following at least 2 hours of treatment with CMP05/CMP05-07. Additionally, it is unclear why the cell viability is consistently no better than ~80% even at extremely low doses of the small molecule drugs in Fig. 2b.

We apologize for the lack of clarity about the ~80% figure demonstrated for viability at very low doses. In Fig. 2b we quantify cell health by measurement of nucleus size and morphology (where small condensed nuclei represent cellular toxicity). Similarly, cells undergoing division can be detected in this manner. The baseline can vary slightly dependent upon where the size/intensity cut-off is established between condensed/uncondensed cells. In these experiments, the 80% represents the baseline measurement as also observed in the DMSO control samples (leftmost graph). It is essentially an arbitrary cutoff, rather than an absolute measurement of viability.

To address this, we have now represented cellular health after compound treatment as a percentage of the DMSO control and removed the DMSO graphs from the figure. This does not alter the conclusions drawn in the figure and we hope this makes the figure easier to interpret.

Unfortunately, there is not a clear approach for us to assess cell toxicity after 24 hours. The main reason is that the media change step required in our treatment removes dead/damaged cells. For example, this means a WST-1 assay cannot reliably quantitate proliferation between conditions. In this instance, we believe the microscopic approach is the best-suited method of estimating compound-induced toxicity. Also, as LDH is meant to measure acute toxicity via plasma membrane perturbation, we believe it is a fitting use to use it directly after the 2-hour exposure to our compounds rather than at a 24-hr timepoint.

2. Proton Sponge. The claims regarding proton sponge as the mechanism for disruption are a bit circumstantial. Others have shown that active membrane disruption reagents (and not just proton sponge systems) can cause endosome enlargement. Some thoughts that could be considered are a pH-dependent hemolysis or synthetic vesicle disruption assay to

Postal address

Karolinska Institutet, Dept. of Laboratory Medicine,
Clinical Research Center (CRC), Novum
Hälsövägen 7
SE-141 52 Huddinge, Sweden

Org. number 202100 2973

Visiting address

Clinical Research Center (CRC)
CRC-Novum Science Park
5th floor, Blickagången 6,
Flemingsberg - Huddinge

Contact

Direct +46 (0)8 585 838 73
Lab +46 (0)8 585 836 53
Fax +46 (0)8 585 836 50
E-mail jeremy.bost@ki.se

more directly determine that the molecules do not have any active membrane disruption activity. It seems unlikely that small molecules would but may be necessary to solidify the conclusion. Though it doesn't separate out membrane active vs proton sponge mechanisms of endosome disruption Bafilomycin A can be an additional nice tool for proving dependency of molecule activity on the endosomal acidifying proton pump (<https://doi.org/10.1038/s41467-019-12906-y>).

We agree with the reviewer that the claim of the proton sponge effect as the mechanism for disruption may be circumstantial. To this end, we have conducted the suggested Bafilomycin experiment and it has been included in the manuscript as a new supplementary figure 5. The bafilomycin experiment supports our theory that the endosomolytic activity of the compounds relies on endosomal acidification. As the reviewer points out, we can only claim that the mechanism may be dependent on endosomal acidification, but we cannot claim the proton-sponge effect is proven. Therefore, we have removed mention of the proton-sponge effect in our discussion. We have also cited the recommended article in the results.

Minor Comments:

1. Greater elaboration on the intended application of these new compounds should be discussed. Is this intended to primarily serve as an *in vitro* transfection enhancer? If so, it would strengthen the manuscript to consider demonstrating with other types of biological cargos. If therapeutic potential is the goal, it would be desirable to demonstrate at least *in vitro* splice switching of a disease-relevant target vs solely a luc reporter. CMP01 has been previously demonstrated in an animal model for example (reference 18).

This is a great point raised by the reviewer and our ongoing work is to determine more applications for these compounds. Similarly to our comment to reviewer 1, we can say that these compounds effectively transfect other biological compounds (cas9 + sgRNA) *in vitro*, but not all others (naked mRNA).

We have attempted *in vivo* experiment targeting the dystrophin gene, but we were limited in our experimental set-up by time and cost (we only had permission to use wt mice). Unfortunately, this round of *in vivo* experiments was unsuccessful. Our ongoing work revolves around using the compounds to bolster other delivery strategies, such incorporating them into various nanoparticles or using them for delivery of other genetic tools *ex vivo*. As this manuscript concerns delivery of oligonucleotides, we choose to refrain from incorporating other delivery cargo here.

2. The authors should consider citing the first paper where Galectin recruitment imaging was utilized for library screening for endosome escape. <https://doi.org/10.1021/acsnano.8b05482>

We fully agree, and the relevant paper has been added.

3. In Figure 1b-c, please clarify the statements that “data comes from n=3 experiments” – does this mean that the experiment was repeated three times, or that a single experiment was conducted with n=3 biological replicates?

We agree this is worded incorrectly and has been changed to state “data comes from N=3 experiments, each with n=3 biological replicates.”

4. Statistically significant differences by two-way ANOVA should be shown in Figure 1c. If none of the differences is significant, then this should be indicated the caption.

We fully agree that there should be a significance test in this figure. We have now included this in the manuscript.

5. For line 252/Figure 4, please explain why HeLa and HuH7 cells were treated with different concentrations of CMP05, and notably why HeLa cells received a higher dose when they appear to be the more susceptible cell line to toxicity from CMP05.

The reviewer is indeed correct that slightly different concentrations were used in Fig.4, however, we have not observed that HeLa is more susceptible to toxicity than Huh7. We have updated Figure 2 to more clearly relay the tox data - the toxicity seen in HeLa at 5uM is not critical at the 2 hr timepoint. Also, the Huh7 seemed reluctant sometimes to adhere on the glass surface required for the confocal imaging, so we may have used 2.5uM to ensure they were happy enough to adhere well.

6. Please add information about temperature/environmental control for timelapse experiments (e.g. Figure 6) to the methods section. If these experiments were conducted at room temperature, please address whether this could impact the kinetics of the observed phenomena.

Postal address

Karolinska Institutet, Dept. of Laboratory Medicine,
Clinical Research Center (CRC), Novum
Hälsövägen 7
SE-141 52 Huddinge, Sweden

Org. number 202100 2973

Visiting address

Clinical Research Center (CRC)
CRC-Novum Science Park
5th floor, Blickagången 6,
Flemingsberg - Huddinge

Contact

Direct +46 (0)8 585 838 73
Lab +46 (0)8 585 836 53
Fax +46 (0)8 585 836 50
E-mail jeremy.bost@ki.se

We apologize for this omission of details regarding time-lapse experiments. Samples were kept within a humidified environmental chamber during imaging, maintained at 37°C and supplemented with 5% CO₂ as per standard growth conditions. The kinetics observed are therefore not expected to be impacted. We have updated the methods section appropriately:

“Dose-response curves of indicated compounds were dispensed by utilizing an Echo 655T acoustic dispenser (Labcyte) into source plates (Greiner, cat no. 781280) containing growth media. At experimental start points, media containing appropriate compounds and doses was transferred to plates using a liquid handling robot (Agilent Bravo).

At assay endpoints, cells were washed 2X PBS at RT and fixed in 4% PFA (VWR, cat no. 9713.1000) for 15 min at RT. Cells were washed a further 3X PBS before the addition of PBS + 1 µg/ml Hoechst 33342 (ThermoFisher Scientific, cat no. H21492) for a minimum of 1 hr before imaging. For time-lapse experiments, cells were imaged within a humidified environmental chamber that was maintained at 37°C and supplemented with 5% CO₂.”

Postal address

Karolinska Institutet, Dept. of Laboratory Medicine,
Clinical Research Center (CRC), Novum
Hälsövägen 7
SE-141 52 Huddinge, Sweden
Org. number 202100 2973

Visiting address

Clinical Research Center (CRC)
CRC-Novum Science Park
5th floor, Blickagången 6,
Flemingsberg - Huddinge

Contact

Direct +46 (0)8 585 838 73
Lab +46 (0)8 585 836 53
Fax +46 (0)8 585 836 50
E-mail jeremy.bost@ki.se

REVIEWERS' COMMENTS:

Reviewer #1 (Remarks to the Author):

I recommend acceptance. All my concerns have been addressed.

Reviewer #2 (Remarks to the Author):

The authors nicely addressed the comments.